# CCCP is Frank-Wolfe in disguise

**Alp Yurtsever**
Umeå University
alp.yurtsever@umu.se

**Suvrit Sra**
Massachusetts Institute of Technology
suvrit@mit.edu

## Abstract

This paper uncovers a simple but rather surprising connection: it shows that the well-known convex-concave procedure (CCCP) and its generalization to constrained problems are both special cases of the Frank-Wolfe (FW) method. This connection not only provides insight of deep (in our opinion) pedagogical value, but also transfers the recently discovered convergence theory of nonconvex Frank-Wolfe methods immediately to CCCP, closing a long-standing gap in its *non-asymptotic* convergence theory. We hope the viewpoint uncovered by this paper spurs the transfer of other advances made for FW to both CCCP and its generalizations.

## 1 Introduction

We study non-convex *difference of convex (DC)* optimization problems of the form:

$$\min_x \quad f(x) - g(x), \quad x \in \mathcal{D}, \tag{1.1}$$

where both $f$ and $g$ are convex functions and $\mathcal{D}$ is a constraint set that might itself be non-convex (we will specify its structure later). Throughout, we assume that the optimal value $f^\star - g^\star$ is finite.

Formulation (1.1) is a DC program. When $f$ and $g$ are smooth, a powerful local method is Convex-Concave Procedure (CCCP) [Yuille and Rangarajan, 2003] that reduces (1.1) to a sequence of convex problems. CCCP is not only algorithmically appealing, it is also widely applicable and numerous algorithms can be viewed as special cases of CCCP, most notably the famous EM algorithm—see [§4 Yuille and Rangarajan, 2003].

But despite its wide applicability (see §2.2) and elegance, convergence theory for CCCP is quite limited. Its *asymptotic* convergence can be obtained by viewing it as a specialization of the DC algorithm (DCA) [Le Thi and Pham Dinh, 2018], or via a more refined analysis designed for CCCP in [Lanckriet and Sriperumbudur, 2009]. A long-standing gap in the literature on CCCP has been to obtain *non-asymptotic* convergence theory that establishes convergence to an $\varepsilon$-stationary point in $O(\text{poly}(1/\varepsilon))$ or fewer iterations.[1]

The starting point of this paper is a discovery that is elementary yet surprising: *CCCP is a special case of the Frank-Wolfe (FW) method*! This discovery not only provides insight of deep (in our opinion) pedagogical value, but it also transfers the convergence theory of nonconvex FW to CCCP, which closes the abovementioned gap in CCCP's non-asymptotic convergence theory.

### 1.1 Main contributions

In light of this short background, we summarize now the key contributions of this paper.

- We recognize the basic (unconstrained) CCCP method to be a special case of Frank-Wolfe. By the same argument, even its generalization to convex constrained instances of (1.1) is a special case of FW. This realization allows us to immediately transfer non-asymptotic convergence theory of FW to CCCP and convex constrained CCCP.

---

[1]We became aware of some recent discoveries on the non-asymptotic convergence of basic CCCP in [Abbaszadehpeivasti et al., 2021, Lê-Huu and Alahari, 2021] after completing this work—see Remark 3.4.

36th Conference on Neural Information Processing Systems (NeurIPS 2022).

- Building on the above connection, we subsequently propose a new variant of FW (called FW+) that applies to the most general form of CCCP (called CCCP+), where the constraint set $\mathcal{D}$ in (1.1) is specified via several DC constraints. The variant FW+ not only allows us to syntactically view CCCP+ as a special case, but more importantly allows us to transfer non-asymptotic convergence guarantees that we prove for FW+ directly to CCCP+.

While the FW+ variant and its analysis are slightly more subtle than basic FW, the connection between CCCP and FW uncovered by our work is remarkably simple. The introduction of FW+ itself is an immediate consequence of the connection uncovered. It is also quite plausible that the connections described in this paper will have various further consequences, including development of new variants of CCCP and other related methods. For a more concrete discussion of potential implications, see the discussion in §5.

Finally, we note a few additional connections of the view uncovered in this paper in the appendix. In particular, we show that various other methods such as the proximal point method, mirror descent and mirror prox can also be seen as special instances of the FW method. A deeper exploration of these connections is left as future work.

## 1.2 Related work

**FW.** Frank and Wolfe [1956] proposed the FW algorithm originally for minimizing a convex quadratic function over a polyhedron and proved $\mathcal{O}(k^{-1})$ convergence rate in this setting. Levitin and Polyak [1966] extended the analysis and proved the same rate for minimization of an arbitrary smooth convex function over a generic convex compact set, and Canon and Cullum [1968] showed that this rate is optimal. The analysis is extended for a broader Hölder-smooth function class by Nesterov [2018]. Jaggi [2013] presented an affine invariant analysis for the FW algorithm by replacing smoothness with the bounded curvature assumption, see (2.3). The analysis of FW for non-convex functions appeared in [Lacoste-Julien, 2016]—the method finds a stationary point as the gap function converges to zero with $\mathcal{O}(k^{-1/2})$ rate, see (2.4) for the gap function. It is easy to show that the rate improves to $\mathcal{O}(k^{-1})$ for *minimization of a concave function*, see Lemma 2.1. An application of the FW algorithm to minimization of a concave function appears in [Luss and Teboulle, 2013].

Design variants of FW with away, pairwise, or in-face steps and line-search or momentum strategies; and extensions of FW for stochastic gradients or block-coordinate updates are extensively studied in the literature. We refer to [Kerdreux, 2020] for an excellent overview on these recent developments. To our knowledge, the variant FW+ introduced in this paper is new.

**CCCP.** The CCCP method of Yuille and Rangarajan [2003] is a special case of the more general DC algorithm (DCA) [Tao, 1997, Le Thi and Pham Dinh, 2018]. CCCP often provides a strong baseline method for tackling non-convex problems where a DC structure is already known or relatively easy to obtain— see [Lipp and Boyd, 2016] for some recent variations on the CCCP method. As already noted, asymptotic convergence of CCCP follows from the more general convergence theory of DCA [Tao, 1997], while a simplified, more direct convergence analysis was obtained in [Lanckriet and Sriperumbudur, 2009], who analyzed convergence of both function values and iterates; moreover, these convergence results apply to the most general CCCP+ formulation (DC objective with DC constraints). A more recent work [Khamaru and Wainwright, 2018] considers gradient and subgradient based alternatives to DCA and CCCP, establishing their corresponding non-asymptotic convergence; however, its focus is on alternatives and its guarantees do not transfer to CCCP, and thus *a fortiori* not to CCCP+. Both CCCP and DCA have received a large number of applications, in a variety of areas including statistics, machine learning, signal processing, operations research, and many more—we refer the reader to the extensive list documented in the survey [Le Thi and Pham Dinh, 2018] and to examples in [Yuille and Rangarajan, 2003].

We became aware of some important items of related work in the recent literature after completing this work. [Abbaszadehpeivasti et al., 2021] proposes non-asymptotic convergence guarantees for unconstrained DC optimization. Their analysis is based on the performance estimation technique [Drori and Teboulle, 2014], which is a computer-assisted methodology for estimating the worst-case complexity of an algorithm by solving semidefinite programs. They conjecture the convergence guarantees through systematic numerical experimentation and then analytically verify them. Their proof provides limited insight and does not extend to the more general constrained settings.

The closest in relevance to our work is [Lê-Huu and Alahari, 2021]. The authors therein recognize the connection between CCCP and FW from a conceptually different viewpoint and show that the unconstrained CCCP is a special instance of the *generalized* FW studied in [Bredies et al., 2009]. Generalized FW is an extension of FW for solving composite optimization problems through partial linearization leading to a modified FW subproblem. Although not identified explicitly before, generalized FW is algebraically equivalent to the original FW applied to an epigraph reformulation in the context of CCCP. We independently discovered the connection between CCCP and FW through the lens of this epigraph reformulation.

We emphasize the value of our conceptual perspective: the epigraph reformulation allows us to transfer existing guarantees of FW to CCCP *without having to perform new convergence analysis* for the unconstrained and convex constrained CCCP case. We further expound this connection and develop it to the DC constrained setting for CCCP+, leading to the discovery of FW+ as a natural extension of the FW method for functional constraints. Our FW+ analysis gives the first non-asymptotic convergence guarantees for the CCCP+. We further observe that the proximal point method (PPM), mirror descent, and mirror prox are also special cases of FW applied to a suitable epigraph reformulation, see Appendices A and B. Since we reduce PPM to FW, and many other optimization methods reduce to PPM, all these methods are inherently special instances of FW.

## 2 Background on Frank-Wolfe and CCCP

### 2.1 Frank-Wolfe Algorithm

We begin by recalling the basic Frank-Wolfe method (FW) and some of its key properties. We focus in particular on the nonconvex case, as it is key to the rest of our theory. Consider therefore the prototypical constrained optimization template:

$$\min_{\omega} \ \phi(\omega) \quad \text{s.t.} \quad \omega \in \mathcal{D}, \tag{2.1}$$

where $\mathcal{D}$ is a closed and convex set in $\mathbb{R}^n$ and $\phi$ is a continuously differentiable (potentially nonconvex) function over the domain $\mathcal{D}$. FW applied to (2.1) performs the following steps for $k = 1, \dots$

$$\omega_k^* \in \operatorname*{argmin}_{\omega \in \mathcal{D}} \ \phi(\omega_k) + \langle \nabla\phi(\omega_k), \, \omega - \omega_k \rangle,$$
$$\omega_{k+1} = (1 - \eta_k)\omega_k + \eta_k \omega_k^*, \tag{FW}$$

where $\eta_k \in [0, 1]$ is a step-size. There are multiple design choices for the step-size. A common choice is the greedy step-size

$$\eta_k = \operatorname*{argmin}_{\eta \in [0,1]} \ \phi\big((1 - \eta)\omega_k + \eta\omega_k^*\big). \tag{2.2}$$

The standard convergence theory of FW crucially depends on the boundedness of *curvature constant* of $\phi$ over $\mathcal{D}$, defined as follows:

$$C_\phi := \sup_{\substack{\omega, \hat{\omega} \in \mathcal{D}; \ \eta \in [0,1] \\ \bar{\omega} = (1-\eta)\omega + \eta\hat{\omega}}} \frac{2}{\eta^2}\Big(\phi(\bar{\omega}) - \phi(\omega) - \langle \nabla\phi(\omega), \, \bar{\omega} - \omega \rangle\Big). \tag{2.3}$$

The bounded curvature assumption is closely related to the Lipschitz smoothness assumption. If $\nabla\phi$ is $L$-Lipschitz continuous and the domain $\mathcal{D}$ has a bounded diameter $D$, then $C_\phi \leq LD^2$. However, the former is more general. For instance, it is easy to see that $C_\phi = 0$ if $\phi$ is concave. This becomes useful in our analysis later in this paper, and allows us to derive guarantees without any Lipschitz smoothness assumption on the functions involved and without an explicit assumption on the boundedness of the domain.

If $\phi$ is convex and $C_\phi$ is bounded, then the sequence $\{\omega_k\}_{k \geq 0}$ generated by (FW) converges to a solution in objective value (see e.g., [Jaggi, 2013, Theorem 1]); where the convergence rate is

$$\phi(\omega_k) - \phi^\star \leq \frac{2C_\phi}{k+1}, \quad \text{where} \quad \phi^\star := \min_{\omega \in \mathcal{D}} \phi(\omega).$$

When $\phi$ is non-convex, (FW) can be shown to find a stationary point [Lacoste-Julien, 2016, Theorem 1], with the following guarantee: there exists an index $\tau \in [k]$, such that

$$\langle \nabla\phi(\omega_\tau), \, \omega_\tau - \omega \rangle \leq \frac{\max\{C_\phi, \, 2(\phi(\omega_1) - \phi^\star)\}}{\sqrt{k}} \qquad \forall \omega \in \mathcal{D}.$$

This bound is often presented in terms of the so-called FW-gap:

$$\text{gap}(\omega_\tau) := \max_{\omega \in \mathcal{D}} \langle \nabla \phi(\omega_\tau), \omega_\tau - \omega \rangle = \langle \nabla \phi(\omega_\tau), \omega_\tau - \omega_\tau^* \rangle. \qquad (2.4)$$

This quantity naturally arises in the FW algorithm and its analysis, and it provides a reliable stopping criterion.

In our analysis, we particularly need to apply FW to concave objectives. In this case, (FW) exhibits better convergence rates than general non-convex costs. More precisely, we note the following result.

**Lemma 2.1.** *Suppose that the sequence $\{\omega_k\}$ is generated by the FW method with greedy step-size* (2.2) *applied to Problem* (2.1) *with a concave objective. Then, there exists $\tau \in [k]$, such that*

$$\langle \nabla \phi(\omega_\tau), \omega_\tau - \omega \rangle \leq \frac{\phi(\omega_1) - \phi^\star}{k} \qquad \forall \omega \in \mathcal{D}.$$

*Proof.* First, observe that $\eta_k = 1$ in (2.2) when $\phi$ is concave. Then, (FW) becomes

$$\omega_{k+1} \in \underset{\omega}{\text{argmin}} \ \phi(\omega_k) + \langle \nabla \phi(\omega_k), \omega - \omega_k \rangle \quad \text{s.t.} \quad \omega \in \mathcal{D}.$$

By concavity of $\phi$, we have

$$\max_{\omega \in \mathcal{D}} \langle \nabla \phi(\omega_k), \omega_k - \omega \rangle = \langle \nabla \phi(\omega_k), \omega_k - \omega_{k+1} \rangle \leq \phi(\omega_k) - \phi(\omega_{k+1}). \qquad (2.5)$$

Then, averaging (2.5) over $k$ gives

$$\frac{1}{k} \sum_{i=1}^{k} \max_{\omega \in \mathcal{D}} \langle \nabla \phi(\omega_i), \omega_i - \omega \rangle \leq \frac{1}{k} (\phi(\omega_1) - \phi_\star). \qquad (2.6)$$

We conclude the proof by noting that the minimum over $k$ is less than or equal to the average. $\square$

## 2.2 The Convex-Concave Procedure (CCCP)

The starting point for the Convex-Concave Procedure (CCCP) of Yuille and Rangarajan [2003] is the unconstrained DC program

$$\min_x \quad f(x) - g(x), \qquad (2.7)$$

where both $f$ and $g$ are $C^1$ (once continuously differentiable) convex functions; CCCP can also be viewed as a special case of the more general DCA algorithm [Tao, 1997] that does not require differentiability. The key idea behind the success of CCCP is to use the convexity of $g(x)$ to linearize it and obtain the following global upper bound on $f(x) - g(x)$:

$$Q(x; y) := f(x) - g(y) - \langle \nabla g(y), x - y \rangle. \qquad (2.8)$$

At each iteration, CCCP then updates its guess by solving the convex problem

$$x_{k+1} \in \underset{x}{\text{argmin}} \ Q(x; x_k). \qquad (2.9)$$

A solution to (2.9) always exists because $f^\star - g^\star$ is finite by assumption and

$$\min_x Q(x; x_k) \geq \min_x f(x) - g(x) = f^\star - g^\star. \qquad (2.10)$$

CCCP is thus a specific Majorize-Minimize (MM) algorithm [Hunter and Lange, 2004], and owing to the update (2.9), it generates a monotonically decreasing sequence $\{f(x_k) - g(x_k)\}_{k \geq 0}$ of objective values. When $f$ is differentiable, update (2.9) is tantamount to the implicit update

$$\nabla f(x_{k+1}) = \nabla g(x_k). \qquad (\text{CCCP})$$

Note that the subproblem (2.9) is a convex optimization problem that *does not* approximate the function $f(x)$, and thus likely offers a tighter approximate model to the DC cost (2.7), an aspect that might help explain its strong empirical performance [Lipp and Boyd, 2016].

The original CCCP paper [Yuille and Rangarajan, 2003] presented various important applications that could be viewed through the CCCP lens, of which perhaps the EM algorithm [Dempster et al., 1977] and Sinkhorn's method for matrix scaling [Sinkhorn, 1967] two very familiar special cases. A formal study of CCCP's asymptotic convergence properties was undertaken in [Lanckriet and Sriperumbudur, 2009], who studied all three variants: unconstrained CCCP, convex constrained, as well DC constrained CCCP.

# 3 Unconstrained CCCP and convex constrained CCCP via FW

Our first result shows that CCCP can be viewed as a special case of the FW method.

**Proposition 3.1.** *CCCP is equivalent to the FW method applied to the following epigraph reformulation of Problem* (2.7):

$$\min_{x,t} \quad t - g(x) \quad \text{s.t.} \quad f(x) \leq t. \tag{3.1}$$

*Proof.* If we apply FW to (3.1), the linear minimization step is

$$(x_k^*, \, t_k^*) \quad \in \quad \arg\min_{(x,t)} \quad t - \langle \nabla g(x_k), \, x \rangle \quad \text{s.t.} \quad f(x) \leq t. \tag{3.2}$$

KKT conditions are necessary and sufficient for optimality in Problem (3.2) since it is a convex optimization problem and the Slater's condition holds. The Lagrangian of this problem is

$$\mathcal{L}(x, t, \lambda) = t - \langle \nabla g(x_k), \, x \rangle + \lambda(f(x) - t).$$

Lagrangian stationarity condition implies

$$\left. \begin{aligned} \nabla_x \mathcal{L}(x, t, \lambda^*) \big|_{x = x_k^*, \, t = t_k^*} &= -\nabla g(x_k) + \lambda^* \nabla f(x_k^*) = 0 \\ \nabla_t \mathcal{L}(x, t, \lambda^*) \big|_{x = x_k^*, \, t = t_k^*} &= 1 - \lambda^* = 0 \end{aligned} \right\} \quad \nabla f(x_k^*) = \nabla g(x_k). \tag{3.3}$$

We also get $t_k^* = f(x_k^*)$ by the complementary slackness condition.

Finally, FW updates its estimate via

$$x_{k+1} = (1 - \eta_k) x_k + \eta_k x_k^*, \quad t_{k+1} = (1 - \eta_k) t_k + \eta_k t_k^*,$$

where $\eta_k \in [0, 1]$ is chosen to minimize the objective function in (3.1). Since this function is concave, $\eta_k = 1$, and we conclude that the linear system in (3.3) is exactly the CCCP subproblem. $\square$

Proposition 3.1 recognizes a simple derivation of the CCCP update – reformulate the unconstrained DC program as a concave minimization over a convex set and apply FW. This recognition transfers the convergence theory of FW to CCCP as we present in the next corollary.

**Corollary 3.2.** *Suppose that the sequence $\{x_k\}$ is generated by* (CCCP) *applied to Problem* (2.7). *Then, there exists $\tau \in [k]$, such that*

$$f(x_\tau) - f(x) - \langle \nabla g(x_\tau), \, x_\tau - x \rangle \; \leq \; \frac{1}{k} \big( f(x_1) - g(x_1) - (f^* - g^*) \big) \quad \textit{for all } x.$$

*Proof.* By Proposition 3.1, CCCP is equivalent to FW applied to (3.1). Observe that (3.1) is an instance of (2.1) with $\omega = (x, t)$, $\phi(\omega) = t - g(x)$, and $\mathcal{D} = \{(x, t) : f(x) \leq t\}$. Since $\phi$ is concave, we can use Lemma 2.1. Hence, there exists $\tau \in [k]$, such that

$$t_\tau - t - \langle \nabla g(x_\tau), \, x_\tau - x \rangle \leq \frac{(t_1 - g(x_1)) - (t^\star - g^\star)}{k} \qquad \forall (x, t) : f(x) \leq t.$$

$t^\star = f^\star$ by definition, $t_1 = f(x_1)$ by initialization, and $t_\tau = f(x_\tau)$ by complementary slackness, see the proof of Proposition 3.1. $\square$

Corollary 3.2 establishes non-asymptotic convergence guarantees of CCCP to a stationary point of the unconstrained DC program (2.7). Remarkably, this result does not require any assumptions on the Lipschitz-smoothness of the functions or an explicit assumption on the boundedness of the domain.

The main result in Corollary 3.2 is based on the variational inequality characterization of first-order stationarity. Similar measures are used in the literature for various problems [Yurtsever et al., 2021]. The next lemma clarifies the value of this stationarity condition.

**Lemma 3.3.** *A point $x^*$ is a first-order stationary point of Problem* (2.7) *if*

$$f(x^*) - f(x) - \langle \nabla g(x^*), \, x^* - x \rangle \; \leq \; 0 \quad \textit{for all } x. \tag{3.4}$$

*Proof.* By definition, the point $x^*$ is first-order stationary if

$$\langle \nabla f(x^*) - \nabla g(x^*), \, x^* - x \rangle \leq 0 \quad \text{for all } x. \tag{3.5}$$

Since $f$ is convex, $f(x^*) - f(x) \leq \langle \nabla f(x^*), \, x^* - x \rangle$, and (3.5) immediately implies (3.4).

Next, we show (3.5) implies (3.4) as well. Suppose (3.4) holds and consider $x = x^* + \alpha s$ for an arbitrary direction $s$ and step-length $\alpha > 0$:

$$f(x^* + \alpha s) - f(x^*) - \langle \nabla g(x^*), \, \alpha s \rangle \, \geq \, 0.$$

We divide both sides by $\alpha$ and take the limit as $\alpha \to 0^+$:

$$\lim_{\alpha \to 0^+} \frac{f(x^* + \alpha s) - f(x^*)}{\alpha} - \langle \nabla g(x^*), \, s \rangle = \langle \nabla f(x^*), \, s \rangle - \langle \nabla g(x^*), \, s \rangle \, \geq \, 0.$$

Finally, we arrive at (3.5) by substituting $s = \alpha^{-1}(x - x^*)$.

In conclusion, (3.4) and (3.5) are equivalent for Problem (2.7). $\qquad\square$

**Remark 3.4** (Comparison with the existing results)**.** There are two other recent works that provides non-asymptotic convergence guarantees for CCCP in the unconstrained setting.

1. Abbaszadehpeivasti et al. [2021] present two main results:

(i) Under the assumption that at least one of the terms, $f$ or $g$ is smooth, they show

$$\min_{\tau \in [k]} \|v_\tau - u_\tau\| \leq \mathcal{O}(1/\sqrt{k}) \quad \text{where} \quad v_\tau \in \partial f(x_\tau) \text{ and } u_\tau \in \partial g(x_\tau).$$

(ii) For the non-smooth setting, they show

$$\min_{\tau \in [k]} \max_{x} \left\{ f(x_\tau) - f(x) - \langle u_\tau, \, x_\tau - x \rangle \right\} \leq \mathcal{O}(1/k) \quad \text{where} \quad u_\tau \in \partial g(x_\tau).$$

This second result matches our guarantees in Corollary 3.2. As we also note above, Corollary 3.2 does not require any assumption on the smoothness, and it holds with subgradients as well.

2. Lê-Huu and Alahari [2021] provide convergence guarantees for the generalized FW algorithm [Bredies et al., 2009] in the non-convex settings. Moreover, they identify CCCP as a special instance of generalized FW. As a result, the guarantees that they present for the 'concave $f$' setting (which corresponds to the DC template) in Theorem 1 in their paper is equivalent to our Corollary 3.2.

Intriguingly, in this setting, the generalized FW algorithm itself is algebraically equivalent to the standard FW applied to the epigraph reformulation (3.1). This recognition, although it seems simple, is interesting on its own right because the recent literature on FW habitually associates the algorithm with optimization on convex, closed and *bounded* domains. Nevertheless, the bounded domain assumption can be relaxed without losing the guarantees, as

$$\forall \hat{\omega} \in \mathcal{D}, \quad \min_{\omega \in \mathcal{D}} \langle \nabla \phi(\hat{\omega}), \, \omega \rangle \quad \text{is bounded below.} \tag{3.6}$$

In other words, the linear minimization subproblem is well-defined everywhere in the domain. As an interesting fact, Frank and Wolfe [1956] use assumption (3.6) in their original work and not restrict their algorithm for bounded domains, see Hypothesis A in their paper. Assumption (3.6) trivially holds if $\mathcal{D}$ is bounded but not vice versa. It is easy to show that assumption (3.6) is satisfied in Proposition 3.1 since

$$\min_{f(x) \leq t} t - \langle \nabla g(\hat{x}), \, x \rangle \geq \min_{x} \, f(x) - \langle \nabla g(\hat{x}), \, x \rangle$$

$$\geq \min_{x} \, f(x) - g(x) + g(\hat{x}) + \langle \nabla g(\hat{x}), \, \hat{x} \rangle$$

$$\geq f^\star - g^\star + g(\hat{x}) + \langle \nabla g(\hat{x}), \, \hat{x} \rangle$$

and $f^\star - g^\star$ is finite by the blanket assumption.

### 3.1 Convex constrained CCCP via FW

Here the problem of interest is:

$$\min_x \quad f(x) - g(x), \quad \text{s.t.} \quad x \in \mathcal{D}, \tag{3.7}$$

for a closed convex set $\mathcal{D}$. We can equivalently write this problem as

$$\min_x \quad f(x) + \chi_{\mathcal{D}}(x) - g(x), \tag{3.8}$$

where $\chi_{\mathcal{D}}(\cdot)$ denotes the indicator function of $\mathcal{D}$. Since $h := f + \chi_{\mathcal{D}}$ is convex, we can apply the same idea as for the unconstrained case by looking at the equivalent problem

$$\min_{x,t} \quad t - g(x), \quad \text{s.t.} \quad h(x) \leq t. \tag{3.9}$$

If we apply FW to (3.9), the linear minimization step involves solving

$$\min_{x,t} \quad t - \langle \nabla g(x_k), x \rangle, \quad \text{s.t.} \quad f(x) + \chi_{\mathcal{D}}(x) \leq t, \tag{3.10}$$

which is clearly equivalent to solving

$$\min_x \quad f(x) - \langle \nabla g(x_k), x \rangle, \quad \text{s.t.} \quad x \in \mathcal{D}. \tag{3.11}$$

Subproblem (3.11) is precisely the update of convex constrained CCCP.

## 4 CCCP with difference of convex constraints via FW

At this point, it is natural to wonder whether the most general version of CCCP also admits a natural FW interpretation? Consider therefore the DC programming problem with DC constraints:

$$\min_x \quad f_0(x) - g_0(x) \quad \text{s.t.} \quad f_i(x) - g_i(x) \leq 0, \ i = 1, \ldots, m. \tag{4.1}$$

Motivated by the applications in Gaussian processes and support vector machines with missing variables, the *advanced* CCCP for Problem (4.1) is introduced in [Smola et al., 2005, Theorem 1]. This method updates its estimation to a solution of the following subproblem:

$$\begin{aligned} \min_x \quad & f_0(x) - g_0(x_k) - \langle \nabla g_0(x_k), x - x_k \rangle \\ \text{s.t.} \quad & f_i(x) - g_i(x_k) - \langle \nabla g_i(x_k), x - x_k \rangle \leq 0, \qquad i = 1, \ldots, m. \end{aligned} \tag{CCCP+}$$

A large body of DC programming literature focuses on template (4.1), or even more general versions of it. To name a few examples, [Fardad and Jovanović, 2014] studies this template for finding optimal feedback gains in the presence of structural constraints, and [Tao et al., 2016] uses it for solving content-centric sparse multicast beamforming. [Lipp and Boyd, 2016] focuses on the same template and provides a discussion on the applications, also introducing several extensions and variations of CCCP+ (e.g., with line-search). [Shen et al., 2016] proposes a structured way to formulate and handle problems as DC programs with DC constraints, called disciplined convex-concave programming (DCCP). DCCP is a generic framework which covers the boolean linear program, boolean least squares, and quadratic assignment problem as special instances. DC programming has a long history, beyond the references we cite above, we refer to [Le Thi and Pham Dinh, 2018] for an overview.

There is a major challenge to make a natural connection between CCCP+ and FW: The feasible set of (4.1) is non-convex, and the feasible sets for (4.1) and (CCCP+) are not the same. As a result, it seems impossible to write CCCP+ as a special instance of the standard FW method. However, searching for a connection between CCCP+ and FW leads us to the discovery of a more general form of the FW method itself that we call FW+. This method is obtained by linearizing constraints along with the objective function, and it recovers CCCP+ as a special case while remaining amenable to a transparent FW-style convergence analysis.

## 4.1 Extended FW algorithm

Here is the problem template:

$$\min_{\omega \in \mathcal{D}} \quad \phi(\omega) \quad \text{s.t.} \quad \psi_i(\omega) \leq 0, \quad i = 1, \ldots, m, \tag{4.2}$$

where $\mathcal{D}$ is a closed and convex set in $\mathbb{R}^n$ and $\phi$ and $\psi_i$ are continuously differentiable functions with bounded curvature.

The FW+ steps for Problem (4.2) are as follows:

$$\omega_k^* \in \operatorname*{argmin}_{\omega \in \mathcal{D}} \ \phi(\omega_k) + \langle \nabla \phi(\omega_k), \ \omega - \omega_k \rangle,$$
$$\text{s.t.} \quad \psi_i(\omega_k) + \langle \nabla \psi_i(\omega_k), \ \omega - \omega_k \rangle \leq 0, \quad i = 1, \ldots, m \tag{FW+}$$

$$\omega_{k+1} = (1 - \eta_k)\omega_k + \eta_k \omega_k^*.$$

Here, we focus on a setting where both $\phi$ and $\psi$ are concave. In this setting, we can choose $\eta_k = 1$. We defer a more general discussion on FW+ for other settings to an extension of this paper.

**Theorem 4.1.** *Suppose that the sequence $\{\omega_k\}$ is generated by the FW+ algorithm for Problem* (4.2) *with concave functions $\phi$ and $\psi$. Then, there exists an index $\tau \in [k]$, at which $\psi_i(\omega_\tau) \leq 0$ for $i = 1 \ldots, m$, and*

$$\langle \nabla \phi(\omega_\tau), \ \omega_\tau - \omega \rangle \leq \frac{\phi(\omega_1) - \phi^\star}{k}, \quad \forall \omega \in \bigcap_{i=1}^m \{\omega \in \mathcal{D} : \psi_i'(\omega, \omega_\tau) \leq 0\}, \tag{4.3}$$

*where $\psi_i'(\omega, \omega_\tau) := \psi_i(\omega_\tau) + \langle \nabla \psi_i(\omega_\tau), \ \omega - \omega_\tau \rangle$.*

*Proof.* First, we focus on the constraint function $\psi$. By concavity, for all $i$,

$$\psi_i(\omega_{k+1}) \leq \psi_i(\omega_k) + \langle \nabla \psi_i(\omega_k), \ \omega_{k+1} - \omega_k \rangle \leq 0,$$

where the last inequality follows from the constraint in the linear minimization step of FW+.

Similarly, by concavity of $\phi$, we have

$$\langle \nabla \phi(\omega_k), \ \omega_k - \omega_{k+1} \rangle \leq \phi(\omega_k) - \phi(\omega_{k+1}).$$

Averaging this inequality over $k$ gives

$$\frac{1}{k} \sum_{i=1}^k \langle \nabla \phi(\omega_i), \ \omega_i - \omega_{i+1} \rangle \leq \frac{\phi(\omega_1) - \phi(\omega_{k+1})}{k} \leq \frac{\phi(\omega_1) - \phi^\star}{k},$$

where the second inequality holds since $\omega_{k+1}$ is in a restriction of the feasible set of (4.2). Then, there exists an index $\tau \in [k]$, at which

$$\langle \nabla \phi(\omega_\tau), \ \omega_\tau - \omega_{\tau+1} \rangle \leq \frac{\phi(\omega_1) - \phi^\star}{k}.$$

This inequality leads to (4.3) by definition of $\omega_{\tau+1} = \omega_\tau^*$ in (FW+). $\qquad\square$

Before moving on, we characterize stationary points of Problem (4.2) in the next lemma. This gives a clear interpretation of the bounds in (4.1) as a perturbation of the stationarity condition.

**Lemma 4.2.** *Consider Problem* (4.2) *and assume $\phi$ and $\psi$ are concave. Then, $\omega^* \in \mathbb{R}^n$ is a feasible stationary point of Problem* (4.2) *if*

$$\omega^* \in \mathcal{D} \quad \text{and} \quad \psi_i(\omega^*) \leq 0, \quad \forall i = 1, \ldots, m, \tag{feasibility}$$

$$\langle \nabla \phi(\omega^*), \ \omega - \omega^* \rangle \geq 0, \quad \forall \omega \in \bigcap_{i=1}^m \{\omega \in \mathcal{D} : \psi_i'(\omega, \omega^*) \leq 0\}, \tag{stationarity}$$

*where $\psi_i'(\omega, \omega^*) := \psi_i(\omega^*) + \langle \nabla \psi_i(\omega^*), \ \omega - \omega^* \rangle$.*

*Proof.* Let's note that $\psi_i(\omega) \leq \psi_i'(\omega, \omega^*)$ due to concavity, therefore the set considered for the stationarity condition in Lemma 4.2 is a restriction of the feasible set. The lemma implies that this restriction contains all feasible directions at $\omega^*$. Here is a simple proof.

First, suppose that that the conditions in Lemma 4.2 are satisfied at $\omega^*$ and the inequality constraint is active, *i.e.*, $\psi_i(\omega^*) = 0$. Then, $\psi_i'$ is also active since $\psi_i'(\omega^*, \omega^*) = \psi_i(\omega^*) = 0$. Assume that there exists a direction $d \in \mathbb{R}^n$ such that $\bar{\omega} := \omega^* + \alpha d$ is feasible for small enough $\alpha > 0$, and that $d$ is a descent direction, $\langle \nabla \phi(\omega^*), d \rangle < 0$. By the stationarity condition in Lemma 4.2, this implies either $\bar{\omega} \notin \mathcal{D}$ or $\psi_i'(\bar{\omega}, \omega^*) > 0$. The former clearly contradicts our feasibility assumption. Suppose the latter holds, then we get $\langle \nabla \psi_i(\omega^*), d \rangle > 0$, meaning that $d$ is an ascent direction for $\psi_i$ at $\omega^*$. This also contradicts our feasibility assumption – constraint is active and $d$ is an ascent direction, hence $d$ is directed outwards from the feasible region. By contradiction, we conclude that there is no feasible descent direction at $\omega^*$.

Next, suppose the constraint is inactive. This means $\psi'$ is also inactive as $\psi_i'(\omega^*, \omega^*) = \psi_i(\omega^*) < 0$. Then, we can place an open ball $\mathcal{B}(\omega^*)$ of infinitesimal radius centered at $\omega^*$ such that $\psi_i'(\omega, \omega^*) \leq 0$ for all points in $\mathcal{B}(\omega^*)$. This implies $\langle \nabla \phi(\omega^*), \omega - \omega^* \rangle \geq 0$ for all $\omega \in \mathcal{D} \cap \mathcal{B}(\omega^*)$ by the stationarity condition in Lemma 4.2. $\qquad\square$

FW+ is a natural extension of FW with extensively broader applications. Theorem 4.1 and Lemma 4.2 together imply that FW+ provably finds a stationary point of Problem (4.2) under the concavity assumptions with non-asymptotic convergence guarantees. We complement this result in Appendix E by deriving convergence guarantees of FW+ when $\phi$ and $\psi$ are convex. Importantly, FW+ can be used for finding a stationary point of a QCQP (quadratically constrained quadratic program) by solving a sequence of linear programs.

## 4.2   CCCP+ via FW+

We are ready to establish the connection between CCCP+ and FW+ by following similar ideas as in Section 3.

**Proposition 4.3.** *CCCP+ is equivalent to the FW+ method applied to the following epigraph reformulation of Problem* (4.1):

$$
\min_{x, t_0, \ldots, t^m} \quad t_0 - g_0(x) \quad \text{s.t.} \quad f_i(x) \leq t_i, \quad i = 0, 1, \ldots, m \tag{4.4}
$$
$$
g_i(x) \geq t_i, \quad i = 1, \ldots, m.
$$

*Proof.* Problem (4.4) is a special case of (4.2) with

$$
\omega = (x, t_0, \ldots, t_m), \quad \phi(\omega) = t_0 - g_0(x), \quad \psi_i = t_i - g_i(x), \quad \text{and} \quad \mathcal{D} = \{\omega : f_i(x) \leq t_i\}.
$$

We consider FW+ for (4.4). Since $\phi$ and $\psi_i$ are concave, we can use $\eta_k = 1$ and the (FW+) procedure accounts to solving

$$
\min_{x, t_0, \ldots, t_m} \quad t_0 - g_0(x_k) - \langle \nabla g(x_k), x - x_k \rangle
$$
$$
\text{s.t.} \quad f_i(x) \leq t_i, \qquad\qquad\qquad\qquad i = 0, 1, \ldots, m
$$
$$
t_i - g_i(x_k) - \langle \nabla g_i(x_k), x - x_k \rangle \leq 0, \quad i = 1, 2, \ldots, m,
$$

which is clearly equivalent to the (CCCP+) subproblem. $\qquad\square$

This connection also transfers the convergence theory of FW+ to CCCP+.

**Corollary 4.4.** *Suppose that the sequence $\{x_k\}$ is generated by* (CCCP+) *applied to Problem* (4.1)*. Then, there exists an index $\tau \in [k]$, such that*

$$
f_0(x_\tau) - f_0(x) - \langle \nabla g_0(x_\tau), x_\tau - x \rangle \leq \frac{1}{k}\Big((f_0(x_1) - g_0(x_1)) - (f_0^\star - g_0^\star)\Big),
$$

*for all $x$ that satisfies* $\quad f_i(x) - g_i(x_\tau) - \langle \nabla g_i(x_\tau), x - x_\tau \rangle \leq 0, \quad i = 1, 2, \ldots, m.$

*Proof.* By Proposition 4.3 and Theorem 4.1, we get

$$t_{0,\tau} - t_0 - \langle \nabla g_0(x_\tau), \, x_\tau - x \rangle \leq \frac{1}{k} \Big( (t_{0,1} - g_0(x_1)) - (t_0^\star - g_0^\star) \Big)$$

for all $(x, t_0, \ldots, t_m)$ that satisfies

$$f_i(x) \leq t_i, \qquad i = 0, 1, \ldots, m$$
$$t_i - g_i(x_\tau) - \langle \nabla g_i(x_\tau), \, x - x_\tau \rangle \leq 0, \qquad i = 1, 2, \ldots, m.$$

This is equivalent to the desired bounds by eliminating $t_0, t_1, \ldots, t_m$ in these formula. $\qquad \square$

## 5   Implications and Discussion

**Discussion.**   We reiterate an important aspect of the non-asymptotic convergence guarantees proved for CCCP and CCCP+ in this paper. Owing to the special structure involving minimization of concave functions, the convergence rates are ***independent of the curvature*** constant of FW (see (2.3)). This property is valuable since it delivers a convergence analysis of CCCP without imposing usual $L$-smoothness assumptions on either $f$, or $g$ or their difference $f - g$, a limitation that typical first-order methods impose. While surprising at first sight, we can reconcile with this intuitively because CCCP requires solving an optimization subproblem (2.9), which is a stronger oracle than merely a gradient oracle for $f - g$. The ensuing faster convergence (i.e., tighter bound on the rate) may offer one explanation to the intuitive view on why CCCP can often be quite competitive [Yuille and Rangarajan, 2003, Lipp and Boyd, 2016].

**Implications.**   Beyond generic convergence guarantees and the transfer of other progress on FW to CCCP, we note some specific implications worth further study here. The EM algorithm is a special case of CCCP [Yuille and Rangarajan, 2003, §4], and therefore a special case of FW. Thus, one should be able to develop a more refined understanding of global iteration complexity of EM, as well as sharper local convergence properties. Given the large number of settings in machine learning and statistics where the EM algorithm, or more generally, variational methods [Blei et al., 2017] are used, a further deepening of their relation to FW-methods should prove fruitful. Similar implications about algorithms, convergence, and complexity apply to other instances of CCCP that have been studied, notably for problems such as matrix scaling [Sinkhorn, 1967].

**Extensions.**   The connection established between FW and CCCP (FW+ and CCCP+) paves the path for numerous extensions that are worthy pursuing. We hope our work motivates others to pursue some of these or other extensions that these connections can inspire.

The first important extension worthy of study is to obtain stochastic variants of CCCP by building on the recent progress on non-convex stochastic FW [Reddi et al., 2016b] as well as non-convex variance reduced versions in the case of finite-sum problems [Reddi et al., 2016a, Allen-Zhu and Hazan, 2016, Yurtsever et al., 2019].

Another line of research worthy of closer attention is to study the case where either $g$ is non-differentiable, or even both $f$ and $g$ in (1.1) are non-differentiable. In that case, developing an extension of the relation between FW and CCCP to a relation between nonsmooth FW [Thekumparampil et al., 2020] (although, as of now this nonsmooth FW method is limited to convex problems) and DCA should also prove to be fruitful, given the extensive progress DCA has witnessed over the decades [Le Thi and Pham Dinh, 2018]. The case where $f(x)$ is an indicator function of a convex set already fits trivially, as already discussed above. One might also speculate that progress on DCA could be transferred back to discover and develop nonsmooth FW methods.

Finally, given that both DCA and CCCP are special instances of MM methods [Hunter and Lange, 2004], it is possible that other MM methods might be profitably viewed as instances of an optimization procedure (not necessarily FW) whose geometric and convergence properties are better understood. We hope our work inspires the discovery of additional such connections.

## Acknowledgments

Alp Yurtsever received support from the Wallenberg AI, Autonomous Systems and Software Program (WASP) funded by the Knut and Alice Wallenberg Foundation. Suvrit Sra acknowledges support from the NSF CAREER grant (1846088) and the NSF CCF-2112665 (TILOS AI Research Institute).

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
