# Appendix: Additional Implications and Connections

In this appendix we briefly comment on a few additional connections and implications. The main text is self-contained and can be read independent of this appendix.

## A  Proximal Point Method and Frank-Wolfe

Consider the generic unconstrained optimization template

$$\min_x \quad f(x). \tag{A.1}$$

The first connection worth noting is related to the fundamental *proximal point method* (PPM) [Martinet, 1970, Rockafellar, 1976]:

$$x_{k+1} \leftarrow \underset{x}{\operatorname{argmin}} \big(f(x) + \tfrac{1}{2\lambda_k}\|x - x_k\|^2\big). \tag{A.2}$$

Rewrite (A.1) by adding and subtracting a strongly convex function $\phi(x)$:

$$\min_x \quad f(x) + \phi(x) - \phi(x). \tag{A.3}$$

We can apply the idea of epigraph splitting:

$$\min_{x,t} \quad t - \phi(x) \quad \text{s.t.} \quad f(x) + \phi(x) \leq t. \tag{A.4}$$

This is concave minimization over a convex set. Similar to the ideas present in Section 3, application of FW to (A.4) leads to

$$x_{k+1} \leftarrow \underset{x}{\operatorname{argmin}} \quad f(x) + D_\phi(x, x_k), \quad k = 0, 1, \dots$$

which is exactly the Bregman-PPM update. If in particular $\phi(x) = \frac{1}{2}\|x\|^2$, with a suitable step-size choice in FW, we immediately obtain the PPM iteration in (A.1). Notably, in the above discussion $f$ need not be smooth, and we have indeed no loss in generality due to the FW view. Since many other methods in optimization reduce to PPM, and above we have reduced PPM to FW, many other optimization methods also reduce to FW via this view. Exploring implications of such algorithmic reductions is left as a topic for future study.

## B  Mirror Descent and Frank-Wolfe

We revisit (A.3) but this time consider the following epigraph form:

$$\min_{x,t} \quad t + f(x) - \phi(x) \quad \text{s.t.} \quad \phi(x) \leq t. \tag{B.1}$$

The objective is not concave in this case, but we can still use FW with a suitable step-size if the curvature is bounded. The linear minimization step becomes (after eliminating $t$):

$$x_k^* = \underset{x}{\operatorname{argmin}} \quad \phi(x) + \langle \nabla f(x_k) - u_k, x \rangle,$$

where $u_k \in \partial\phi(x_k)$ is a subgradient of $\phi$ at $x_k$. This leads to the inclusion

$$u_k - \nabla f(x_k) \in \partial\phi(x_k^*).$$

From the convex combination step of FW, we have the relation

$$x_k^* = \frac{1}{\eta_k} x_{k+1} + (1 - \frac{1}{\eta_k}) x_k.$$

By combining the last two relations, and rearranging, we get

$$x_{k+1} = \nabla\phi^*\big(u_k - \eta_k \nabla f(x_k)\big), \quad u_k \in \partial\phi(x_k), \tag{B.2}$$

where $\phi^*(y)$ is the Fenchel conjugate of $\phi$, and $\phi^*$ is smooth since $\phi$ is strongly convex. This is exactly the mirror descent update. If in particular $\phi(x) = \frac{L}{2}\|x\|^2$, the objective function in (B.1) becomes concave and we can choose $\eta_k = 1$, which leads to the gradient descent step

$$x_{k+1} = x_k - \frac{1}{L}\nabla f(x_k). \tag{B.3}$$

## B.1 Mirror Prox and Frank-Wolfe

This time, we consider a generic composite optimization problem,

$$\min_x \quad f(x) + g(x). \tag{B.4}$$

Suppose $f$ is $L$-smooth. Then, $f(x) - \frac{L}{2}\|x\|^2$ is concave. Add and subtract $\frac{L}{2}\|x\|^2$ in (B.4):

$$\min_x \quad f(x) - \frac{L}{2}\|x\|^2 + g(x) + \frac{L}{2}\|x\|^2, \tag{B.5}$$

and consider the following epigraph form

$$\min_{x,t} \quad f(x) - \frac{L}{2}\|x\|^2 + t \quad \text{s.t.} \quad g(x) + \frac{L}{2}\|x\|^2 \le t. \tag{B.6}$$

If we apply FW on this formulation, the linear minimization step becomes

$$x_k^* = \operatorname*{argmin}_x \quad \langle \nabla f(x_k) - L \cdot x_k, \, x \rangle + g(x) + \frac{L}{2}\|x\|^2, \tag{B.7}$$

which is exactly the proximal gradient method step

$$x_k^* = \operatorname{prox}_{\frac{1}{L}g}\left(x_k - \frac{1}{L}\nabla f(x_k)\right). \tag{B.8}$$

Since the objective in (B.6) is concave, we can choose unit step-size for FW and get $x_k^* = x_{k+1}$.

This idea trivially extends to the mirror-prox by using a Bregman divergence.

## C  CCCP/FW and Non-Convex Proximal-Splitting

Consider again the basic unconstrained problem $\min_x f(x) - g(x)$. Add and subtract $\frac{1}{2}\|x\|^2$, and then consider the equivalent *dual CCCP problem*

$$(g + \tfrac{1}{2}\|\cdot\|^2)^*(x) - (f + \tfrac{1}{2}\|\cdot\|^2)^*(x). \tag{C.1}$$

When applying CCCP to (C.1) we need to compute the gradient of the second term. This gradient can be obtained by recognizing that $f + \frac{1}{2}\|x\|^2$ is strongly convex, and hence its dual is smooth, so that

$$\nabla(f + \tfrac{1}{2}\|\cdot\|^2)^*(z) = \operatorname*{argmax}_x \langle z, \, x \rangle - f(x) - \tfrac{1}{2}\|x\|^2.$$

But the argmax above is nothing but $\operatorname{prox}_f(z)$. Thus, to apply CCCP to (C.1) we need to solve the subproblem

$$x_k^* \leftarrow \operatorname*{argmin}_x (g + \tfrac{1}{2}\|\cdot\|^2)^*(x) - \langle \operatorname{prox}_f(x_k), \, x \rangle. \tag{C.2}$$

To solve (C.2), we can again compute the dual. As a shorthand, write $h^* \equiv (g + \frac{1}{2}\|\cdot\|^2)^*$. Thus, we obtain that $x_k^* = \operatorname{argmax}\langle \operatorname{prox}_f(x_k), \, x \rangle - h^*(x)$, whereby it must satisfy the optimality condition

$$\operatorname{prox}_f(x_k) = \nabla h^*(x_k^*).$$

From our previous argument (since $h = g + \frac{1}{2}\|\cdot\|^2$), we know that $\nabla h^*(x) = \operatorname{argmax}_z \langle x, z \rangle - g(z) - \frac{1}{2}\|z\|^2$, so that

$$\nabla h^*(x_k^*) = \operatorname{prox}_g(x_k^*) = \operatorname{prox}_f(x_k). \tag{C.3}$$

We state this relation only to highlight it as the prox-analog of the usual implicit update of CCCP, namely, $\nabla f(x_k^*) = \nabla g(x_k)$.

After computing $x_k^*$ via (C.2), or via (C.3), the update step of FW leads to

$$x_{k+1} = (1 - \eta_k)x_k + \eta_k x_k^*, \tag{C.4}$$

Ultimately, the dual view permits us to apply CCCP without assuming differentiability, while obtaining convergence guarantees for it via the FW view applied to the dual problem (C.2).

# D  Frank-Wolfe is a special case of CCCP

We showed in Section 3 that CCCP is a special instance of FW. Interestingly (though unsurprisingly), this connection goes both ways and we can cast FW as an instance of CCCP.

**Example D.1.** Let $f(x)$ be an indicator function of a convex set $\mathcal{X}$. Then,

$$\min_x \; f(x) - g(x) \quad \Leftrightarrow \quad \max g(x), \text{ s.t. } x \in \mathcal{X}.$$

Applying FW to this problem results in the same update, when solving the CCCP subproblem

$$\min_x \; f(x) - g(y) - \langle \nabla g(y), \, x - y \rangle.$$

Since $f$ is an indicator, this problem translates into the linear minimization step of FW

$$x_k^* \in \operatorname*{argmin}_x \; \langle -\nabla g(x_k), \, x \rangle, \quad \text{s.t. } x \in \mathcal{X}.$$

Since the objective is concave, FW can use the unit step-size and $x_{k+1} = x_k^*$.

# E  More details on FW+

In Section 4.1 we analyzed the FW+ method for the important setting where both $\phi$ and $\psi$ are concave. In this appendix, we present additional details on the convergence guarantees of FW+ when $\phi$ and $\psi$ are convex.

**Theorem E.1.** *Suppose that the sequence $\{\omega_k\}$ is generated by the FW+ algorithm with step-size $\eta_k = 2/(k+1)$ applied to Problem (4.2). Suppose $\phi$ and $\psi$ are convex with bounded curvature constants $C_\phi$ and $C_\psi$. Then, the following guarantees hold:*

$$\omega_k \in \mathcal{D}, \quad \phi(\omega_k) - \phi^* \leq \frac{2C_\phi}{k+1}, \quad \text{and} \quad \psi(\omega_k) \leq \frac{2C_\psi}{k+1}. \tag{E.1}$$

*Proof.* $\omega_k$ always remains in $\mathcal{D}$ by design of the algorithm. Next, focus on the constraint function $\psi$. Since $\psi$ has bounded curvature $C_\psi$,

$$\psi(\omega_{k+1}) \leq \psi(\omega_k) + \langle \nabla\psi(\omega_k), \, \omega_{k+1} - \omega_k \rangle + \frac{1}{2}\eta_k^2 C_\psi$$

$$= (1 - \eta_k)\psi(\omega_k) + \eta_k(\psi(\omega_k) + \langle \nabla\psi(\omega_k), \, \omega_k^* - \omega_k \rangle) + \frac{1}{2}\eta_k^2 C_\psi$$

$$\leq (1 - \eta_k)\psi(\omega_k) + \frac{1}{2}\eta_k^2 C_\psi,$$

where the last inequality follows from the constraint in the linear minimization step of FW+. Unrolling this inequality, we obtain

$$\psi(\omega_{k+1}) \leq (1 - \eta_1)\psi(\omega_1) + \frac{1}{2}C_\psi \sum_{i=1}^k \eta_i^2 \prod_{j=i+1}^k (1 - \eta_j).$$

We choose $\eta_k = 2/(k+1)$, hence

$$\psi(\omega_{k+1}) \leq 2C_\psi \sum_{i=1}^k \frac{1}{(i+1)^2} \prod_{j=i+1}^k \frac{j-1}{j+1} = 2C_\psi \sum_{i=1}^k \frac{1}{(i+1)^2} \frac{i(i+1)}{k(k+1)} \leq \frac{2C_\psi}{k+1}.$$

This completes the proof for the constraint function.

The proof of convergence for the objective function follows similarly to the standard proof of the FW method. By the bounded curvature assumption, we have

$$\phi(\omega_{k+1}) - \phi^* \leq \phi(\omega_k) - \phi^* + \langle \nabla\phi(\omega_k), \, \omega_{k+1} - \omega_k \rangle + \frac{1}{2}\eta_k^2 C_\phi$$

$$= \phi(\omega_k) - \phi^* + \eta_k \langle \nabla\phi(\omega_k), \, \omega_k^* - \omega_k \rangle + \frac{1}{2}\eta_k^2 C_\phi.$$

By definition of $w_k^*$,

$$\phi(\omega_{k+1}) - \phi^* \leq \phi(\omega_k) - \phi^* + \eta_k \langle \nabla\phi(\omega_k), \omega - \omega_k \rangle + \frac{1}{2}\eta_k^2 C_\phi,$$
$$\forall \omega \in \{\omega \in \mathcal{D} : \psi(\omega_\tau) + \langle \nabla\psi(\omega_\tau), \omega - \omega_\tau \rangle \leq 0\}.$$

Since $\psi$ is convex, this is a relaxation of the feasible set, it contains $\omega^*$, hence the above inequality holds also for $\omega = \omega^*$:

$$\phi(\omega_{k+1}) - \phi^* \leq \phi(\omega_k) - \phi^* + \eta_k \langle \nabla\phi(\omega_k), \omega^* - \omega_k \rangle + \frac{1}{2}\eta_k^2 C_\phi$$

$$\leq (1 - \eta_k)(\phi(\omega_k) - \phi^*) + \frac{1}{2}\eta_k^2 C_\phi.$$

Then, similar to the above discussion on the convergence of $\psi$, we get

$$\phi(\omega_{k+1}) - \phi^* \leq \frac{2C_\phi}{k+1}.$$

This completes the proof. □