# OpenReview forum: "CCCP is Frank-Wolfe in disguise"
_NeurIPS.cc/2022/Conference — NeurIPS 2022 Accept_

### Official Review · Reviewer_ubjs · 2022-07-11

**Rating:** 3
**Confidence:** 4
**Soundness:** 4 excellent
**Presentation:** 2 fair
**Contribution:** 1 poor

**Summary:**

The authors propose an equivalence between two optimization methods: the convex-concave procedure (CCCP) and the Franke-Wolfe algorithm (FW). This equivalence is shown through an epigraphical lifting, resulting in a concave minimization problem that can be solved by the Frank-Wolfe algorithm with unit step size. This equivalence is shown to hold for unconstrained CCCP, convex constrained CCCP, and difference-of-convex constrained CCCP with various flavors of the FW algorithm with unit step size.

**Questions:**

Besides the FW+/CCCP+ method for difference-of-convex constrained CCCP, what novel contributions does the paper offer?

**Limitations:**

There is discussion of transferring convergence proofs from FW methods to CCCP, for instance there is even mention of possibly using nonsmooth FW to say something about CCCP. However, since all proofs I know of nonsmooth FW variants depend on the compactness of the constraint set, it's not obvious how these would transfer. In this way, the fact that CCCP is equivalent to an unbounded version of FW is itself a limitation that should be addressed in the paper.

**Strengths And Weaknesses:**

I found the idea of the paper to be compelling, since connections of this sort usually offer a lot of insight into design and analysis of algorithms, but there is a question of novelty/originality that I will come back to.

Regarding clarity, it should be stressed that this is not the ordinary FW algorithm, which usually requires the constraint set to be compact and the step size to decay like O(1/k) to prove convergence. Indeed, the constraint set D is not assumed to be bounded in the paper (nor can it be for the equivalence with CCCP to hold in general), yet at some points the diameter of the set D is used (i.e., line 85).

There are some questions about the novelty of the results of the paper, however, when it comes to analyzing FW with unit step size for minimizing a concave function. There is the work [Luss, Teboulle 2012] which analyzes ConGradU, a conditional gradient algorithm with unit stepsize to maximize a convex function over a compact set. The algorithm ConGradU is extremely similar to the algorithm analyzed here, except for the difference in compactness assumption, and should be discussed as a related work.

It also appears that the central idea of the paper, the equivalence between FW and CCCP, has already been published in NeurIPS 2021 [Lê-Huu, Alahari 2021] and was not cited. In their paper, there is a section that describes CCCP as equivalent to the generalized conditional gradient (equivalent to the epigraphical lifting given here) with unit step size. In light of this, it would seem that only the FW+/CCCP+ algorithms for dealing with difference-of-convex constrained CCCP problems are novel and, although interesting, do not seem significant enough on their own to justify publication here.

Finally, I think it would be beneficial to include some sort of numerical experiment demonstrating the method FW+/CCCP+. Seeing the algorithm works in practice would improve the significance and quality of the paper as well.

1. Regularized Frank-Wolfe for Dense CRFs: Generalizing Mean Field and Beyond - Đ.Khuê Lê-Huu, Karteek Alahari, 2021
2. Conditional Gradient Algorithms for Rank-One Matrix Approximations with a Sparsity Constraint - Ronny Luss, Marc Teboulle, 2012

---

> ### Author Response · Authors · 2022-08-02
> **Thank you for your feedback.**
>
> Thank you for your comments. However, there seem to be some key conceptual misunderstandings in the review which we would like to clarify below.
>
> ### Clarity
>
> We actually use the ordinary Frank-Wolfe algorithm (not any specialized modification) for the CCCP-FW connection. The ordinary FW method (as the reviewer also knows) is characterized by two main steps:
> (1) Minimizing the first-order approximation of the objective function (around the current estimate $\omega_k$) over the feasible set (denote the solution of this subproblem by $\omega_k^*$)
> (2) Forming the next estimate ($\omega_{k+1}$) as a convex combination of $\omega_k$ and $\omega_k^*$.
> Decaying step-size $2/(k+1)$ is not mandatory and it is optimal only for deterministic convex objective functions. A global strategy is finding the optimal convex combination by minimizing the objective function on the line segment between $\omega_k$ and $\omega_k^*$. This strategy gives optimal rates both in convex and non-convex settings, see [Jaggi, 2013] and [Julien-Lacoste, 2016]. This strategy gives unit step-size when the objective function is concave (as we explained in lines 137-138 in our draft), hence our choice falls into this global model.
>
> FW analysis **does not** require compactness assumption as long as the subproblems have finite solutions and the curvature is bounded (see the Proof for Theorem 1 in [Jaggi, 2013]). Certainly, the compactness assumption is sufficient for these two conditions to hold. We have these two assumptions implicitly (as noted in our response to the first reviewer; we will make this explicit in our revision), since concavity implies zero curvature (see line 86) and the (CCCP) subproblems are assumed well-defined.
>
> It is not clear why line 85 is a concern. This section presents a background on FW, and we give smooth minimization over compact domain as an example for bounded curvature. Line 86 explicitly states that curvature is zero for concave functions.
>
> ### Novelty and contributions
>
> We do not claim that FW for concave minimization is a novel method. In contrast, we emphasize multiple times in the paper that we are using the standard FW. The analysis in this section is straightforward, and we do not claim contribution on this. Since we do not propose a new algorithm in Section 2, [Luss, Teboulle 2012] is only weakly related. They just happen to use FW on a convex maximization (or concave minimization) problem, but for a specific application with no clear connections to CCCP. We will cite this paper in our revision.
>
> Thank you for bringing [Lê-Huu, Alahari 2021] to our attention. In fact, one of the authors of that work also brought it to our attention recently, and it is truly unfortunate that we were not aware of their work when we submitted ours. However, given its immediate relevance, we will cite it appropriately in our next revision, importantly, with a precise discussion that positions our work vis-a-vis theirs. We respectfully disagree that the results are the same. There are several obvious differences, as well as some subtle ones.
>
> - First of all, [Lê-Huu, Alahari 2021] claim that unconstrained CCCP is a special instance of the *regularized FW*. They emphasize  several times that regularized FW is introduced as an extension of FW (see the first sentence of their abstract) and discuss in Section 3.4 (of their paper) that regularized FW covers standard FW as a special case. In contrast, we formulate CCCP as an instance of the standard FW method applied to an epigraph reformulation of the problem. In fact, their regularized FW itself is an instance of standard FW, and they seem to miss this important conceptual viewpoint. Of course, it is algebraically trivial to see that the two are equivalent, that algebra is not out point: our emphasis is on the **value of the conceptual perspective**:  the epigraph reformulation makes things transparent and allows us to transfer existing guarantees of FW to CCCP (and other settings)--*without having to perform new convergence analysis* for the basic CCCP case. Thanks to the clarity delivered via the epigraph connection, we further observe that proximal point method, mirror descent, and mirror-prox are special cases of FW applied to a suitable epigraph reformulation (see Appendix A and B). Since we reduce PPM to FW, and many other optimization methods reduce to PPM, all these methods inherently reduce to FW.
>
> More importantly, beyond the basic CCCP-FW connection, we present theory for the advanced CCCP setting (that is, a DC program with DC constraints) in Section 4. This extension leads to a new, and rather natural variant of FW that iteratively linearizes both the constraints as well as the objective. To our knowledge, this variant is completely novel to the FW literature. We develop convergence analysis for FW+, which leads to the first non-asymptotic convergence guarantees for the advanced CCCP, a method that is much more general than CCCP.

---

> > ### Author Response · Authors · 2022-08-02
> > **Part 2**
> >
> > Significance of FW+/CCCP+
> > ----
> >
> > This reviewer argues that FW+/CCCP+ is not significant enough. We disagree with this highly subjective comment. Recognizing the value of the pedagogically clean connection between FW/CCCP, expounding it, developing it to obtain a natural extension of FW, while being still able to analyze its non-asymptotic complexity, are in our opinions ideas that deserve to be shared with the wider research community.  Moreover, our discovery of FW+ itself is a proof of the importance of our epigraphical reformulation viewpoint and our conceptual contributions (as well as the multiple reductions of mirror-descent, mirror-prox, PPM, etc. to FW via the epigraphical view as discussed in the appendix.)
> >
> > ### Numerical experiments on FW+
> > The focus of this paper is on the connections between FW/CCCP and FW+/CCCP+. Numerical experiments on FW+ might distract the readers from this focus. We agree that FW+ is worthy of further investigation and we plan it for a follow-up work. There, we will also investigate its practical performance.

---

> > > ### Comment · Reviewer_ubjs · 2022-08-07
> > > **FW+**
> > >
> > > >This reviewer argues that FW+/CCCP+ is not significant enough. We disagree with this highly subjective comment. Recognizing the value of the pedagogically clean connection between FW/CCCP, expounding it, developing it to obtain a natural extension of FW, while being still able to analyze its non-asymptotic complexity, are in our opinions ideas that deserve to be shared with the wider research community. Moreover, our discovery of FW+ itself is a proof of the importance of our epigraphical reformulation viewpoint and our conceptual contributions (as well as the multiple reductions of mirror-descent, mirror-prox, PPM, etc. to FW via the epigraphical view as discussed in the appendix.)
> > >
> > > I think that in it's current form it's not developed enough to be what seems to be the primary novel contribution of the paper, not that the idea itself is necessarily insignificant.
> > >
> > > >The focus of this paper is on the connections between FW/CCCP and FW+/CCCP+. Numerical experiments on FW+ might distract the readers from this focus. We agree that FW+ is worthy of further investigation and we plan it for a follow-up work. There, we will also investigate its practical performance.
> > >
> > > The reason for suggesting experiments in an otherwise theoretical work is because the FW+ algorithm is certainly novel and I think this should be the focus of the paper. Without numerical/practical examples to motivate its study, the proposal of FW+ seems like a generalization for generalization's sake since it is an immediate consequence of the connection between CCCP and FW. Adding interesting examples where this algorithm performs decently could remedy this.

---

> > > > ### Author Response · Authors · 2022-08-09
> > > > **Clarifying FW+ and its context.**
> > > >
> > > > The reviewer raises the point that
> > > >
> > > > > FW+ seems like a generalization for generalization's sake
> > > >
> > > > This statement is completely untrue! It seems surprising to see this point, but in hindsight we now realize that we should have provided additional context behind CCCP+/FW+ --- when we wrote the paper, the conceptual connections were foremost on our minds, so while we did provide several citations, we could have done a better job. We hope the information below helps the reviewer better contextualize our work.
> > > >
> > > > 1. FW+ is a natural extension of FW with extensively broader applications. For instance, FW+ can find a stationary point of a QCQP (quadratically constrained quadratic programming) by solving a sequence of LPs (linear programs). A large body of DC programming literature focuses on our problem template with DC constraints (see Eqn. 4.1 in our draft) or even more general versions of it. DC programming has a long history in optimization; one valuable reference worth looking at is (**Hoai An Le Thi, Tao Pham Dinh, 2018**)
> > > > 2. (**Smola 2005**) introduces CCCP+ motivated by its application in Gaussian Processes and Support Vector Machines with missing variables; this paper is cited in our submission.
> > > > 3. (**Lipp & Boyd, 2015**), also cited in our submission, focus on the same problem template with a discussion on the applications and introduce several extensions and variations of CCCP+ (e.g., line-search).
> > > > 4. (**Shen et al., 2016**) introduce disciplined convex-concave programming, a structured way to formulate and handle problems as DC programs with DC constraints. The template is very broad, including boolean LP, boolean least squares, and quadratic assignment problems as special instances.
> > > > 5. (**Fardad & Jovanovic 2014**) formulate and solve the problem of finding optimal feedback gains in the presence of structural constraints as an instance of this template; and
> > > > 6. (**Tao et al., 2016**) consider this template for solving content-centric sparse multicast beamforming; just to name a few applications.
> > > >
> > > > In summary
> > > > ---
> > > > Our FW+ analysis gives the **first non-asymptotic convergence rate** guarantee for the CCCP+. We will include this discussion on the practical and theoretical implications of CCCP+/FW+ in our revision. This added emphasis is thanks to the feedback from this review process.
> > > >
> > > > We hope that with these updates, we have managed to assuage the reviewer's concerns, and that the reviewer takes a more favorable standpoint on our paper.
> > > >
> > > > References
> > > > ---
> > > >
> > > > - **(Hoai An Le Thi, Tao Pham Dinh, 2018)**. DC programming and DCA: thirty years of developments. https://link.springer.com/article/10.1007/s10107-018-1235-y
> > > > - **(Fardad & Jovanovic, 2014)** On the design of optimal structured and sparse feedback gains via sequential convex programming
> > > > - **(Lipp & Boyd, 2015)**  Variations and extension of the convex–concave procedure
> > > > - **(Shen et al., 2016)** Disciplined Convex-Concave Programming
> > > > - **(Tao et al., 2016)** Content-Centric Sparse Multicast Beamforming for Cache-Enabled Cloud RAN

---

> > ### Comment · Reviewer_ubjs · 2022-08-07
> > **Reponse to authors**
> >
> > ### Clarity
> >
> > It's not that any of the points in the clarity paragraph I wrote are technical flaws in the paper, for instance I understand that FW doesn't require a compact set. Rather, the purpose of the paper is to be pedagogical and a lot of the audience will probably have the problem
> > $\min_{x\in D} f(x)$ with $D$ compact and $f$ convex and lipschitz-smooth, and its associated proofs/stepsizes, in mind when they think of FW.
> >
> > ### Novelty
> >
> > >They just happen to use FW on a convex maximization (or concave minimization) problem, but for a specific application with no clear connections to CCCP. We will cite this paper in our revision.
> >
> > It's not only this, both use unit stepsize as well. Even if the convergence analysis of FW on convex maximization with unit step size isn't meant to be an original contribution of the paper, it's still important to cite related work on the subject (perhaps even more so).
> >
> > >First of all, [Lê-Huu, Alahari 2021] claim that unconstrained CCCP is a special instance of the regularized FW. They emphasize several times that regularized FW is introduced as an extension of FW (see the first sentence of their abstract) and discuss in Section 3.4 (of their paper) that regularized FW covers standard FW as a special case. In contrast, we formulate CCCP as an instance of the standard FW method applied to an epigraph reformulation of the problem. In fact, their regularized FW itself is an instance of standard FW, and they seem to miss this important conceptual viewpoint. Of course, it is algebraically trivial to see that the two are equivalent, that algebra is not out point: our emphasis is on the value of the conceptual perspective: the epigraph reformulation makes things transparent and allows us to transfer existing guarantees of FW to CCCP (and other settings)--without having to perform new convergence analysis for the basic CCCP case. Thanks to the clarity delivered via the epigraph connection, we further observe that proximal point method, mirror descent, and mirror-prox are special cases of FW applied to a suitable epigraph reformulation (see Appendix A and B). Since we reduce PPM to FW, and many other optimization methods reduce to PPM, all these methods inherently reduce to FW.
> >
> > In [Lê-Huu, Alahari 2021] it's written "We conclude that CCCP is a special case of generalized Frank-Wolfe with f concave and unit stepsize" where generalized FW (or generalized conditional gradient) is the same that is used in the well-known [Bredies et al 2009] and also in [Bach 2015]. One can check that regularized FW is not necessary and generalized FW suffices. In the generalized FW setting, one solves
> >
> > $\min_x f(x)+g(x)$
> >
> > using at each iteration the (unconstrained) oracle
> >
> > $\min_x  \langle \nabla f(x_k), x\rangle + g(x)$
> >
> > usually with the otherwise vanilla FW algorithm (depending on assumptions on $g$). When $g$ is the indicator of a convex compact set you recover exactly an instance of vanilla FW.
> >
> > Although [Lê-Huu, Alahari 2021] ultimately studies a different algorithm, which they call regularized FW, the claim about generalized FW doesn't require this. I would say that they analyze a different algorithm but still attain this equivalence as a result, and they note it as such in their paper.
> >
> > Applying vanilla FW to the epigraphical reformulation is equivalent to the generalized FW algorithm. The problem in the generalized FW is $\min_x {f(x)+g(x)}$ which is equivalent to the problem in the epigraphical lifting $\min_{x,t}{f(x) + t}$ subject to $g(x)\leq t$. The step direction found is the same as well. So, the equivalence was definitely published in [Lê-Huu, Alahari 2021].
> >
> > 1. Duality between subgradient and conditional gradient methods - Francis Bach, 2015
> > 2. A Generalized Conditional Gradient Method and its Connection to an Iterative Shrinkage Method - Bredies et al 2009

---

> > > ### Author Response · Authors · 2022-08-09
> > > **Clearing up a misrepresentation**
> > >
> > > We understand and already acknowledge (in our rebuttal) the relevance of [Lê-Huu, Alahari 2021] as well as the algebraical equivalence for the basic CCCP/FW setting. We discovered the CCCP/FW connection completely independently from [Lê-Huu, Alahari 2021], and from a conceptually different viewpoint. We emphasize the importance of this viewpoint; merely because the consequences for the vanilla CCCP case end up being similar does not make our contributions insignificant.
> > >
> > > Importantly, please note that the CCCP/FW connection appears as a note in passing in [Lê-Huu, Alahari 2021]. **Recognizing the value of this connection** and the novel guarantees that it implies, subsequently expounding the connection and developing it to the constrained setting (which leads to a natural extension of the FW method for problems with functional constraints) are important contributions.
> > >
> > > We are saddened to see that the reviewer wishes to downplay these contributions---we are going to be fully acknowledging [Lê-Huu, Alahari 2021] as being the first to recognize the connection between CCCP/FW. _We are neither criticizing their work, nor downplaying it to make ours appear "better"_ --- all that we are trying to do in light of that work is to state the conceptual differences between the two, while pointing out the key contributions that result from our view.
> > >
> > > To provide an analogy:  In his seminal work, [Bach 2015] discovers a duality between FW and dual subgradient methods. We uncover a remarkable set of new connections via the epigraphical splitting viewpoint (i.e., mirror-prox, mirror-descent, and proximal-point-method all can be written as an instance of FW). The discovery of such simple yet fundamental connections is rare in optimization, and in our humble opinion, it deserves to be not dismissed.

---

### Official Review · Reviewer_y2XT · 2022-07-11

**Rating:** 7
**Confidence:** 3
**Soundness:** 4 excellent
**Presentation:** 3 good
**Contribution:** 3 good

**Summary:**

The paper studies the CCCP (convex-concave procedure through the lens of Frank-Wolfe.  Reframing CCCP as Frank Wolfe in epigraph reformulations, the authors provide some of the first non-asymptotic convergence results of CCCP to stationary points.  The work has results for an unconstrained setting, convex constrained setting, and generalizations of FW for DC constraints.

**Questions:**

Convex constrained analysis seems to subsume the unconstrained analysis without too much additional complexity.  Maybe it would be worth combining these?

The acronym MM is used in line 111 without Majorization Minimization.  Including this would help clarity.

**Limitations:**

The authors adequately addressed limitations of the work.

**Strengths And Weaknesses:**

Strengths:
- The paper finds a fundamental and elegant connection between CCCP and FW style algorithms which provide novel convergence bounds.
- EM is a specific case of CCCP.  Further investigation fro insights in this paper may result in improved understanding of convergence of such methods.
-  Paper is well written and contextualized well with related work.  Section F in the appendix does a good job of comparison to the closest related work. the more general analysis, connections to FW, and simpler analysis seem worthwhile even though some finite-time convergence results are known via [Abbaszadehpeivasti et al 2021].

Weakness:
- Notation could be a bit more consistent, for example Theorem 4.1 and Lemma 4.2 could both us $\psi'$
- More discussion on the stationary point notions for CCCP would be useful

---

> ### Author Response · Authors · 2022-08-02
> **Thank you for your feedback.**
>
> Thank you for your valuable comments. We will make sure that the notation is consistent in our next revision. We will also provide a more detailed discussion on the stationarity notions for the CCCP and CCCP+ templates.
>
> ### Combining convex unconstrained and constrained settings
> We agree that the analysis is similar for these two settings, but we decided to focus first on the unconstrained template since it is the most common in the literature. We organized the convex constrained setting as a subsection in section 3 rather than a new section to highlight this similarity. We can further clarify this similarity explicitly in our next revision.
>
> ### Acronym MM is not defined
> Thank you for the notification, we will fix this.

---

### Official Review · Reviewer_cenk · 2022-07-13

**Rating:** 8
**Confidence:** 4
**Soundness:** 4 excellent
**Presentation:** 4 excellent
**Contribution:** 4 excellent

**Summary:**

This paper develops a novel equivalence between the convex-concave procedure (CCCP) for difference of convex (DC) programming and the Frank-Wolfe (FW) algorithm for minimization. The key idea is to show that CCCP is exactly FW applied to an epigraph reformulation of the original DC problem. Based on this equivalence, the authors are able to directly apply computational guarantees for non-convex FW (particularly, FW applied to concave *minimization*) to CCCP, yielding novel non-asymptotic convergence guarantees to a stationary point for CCCP. The equivalence also leads to several natural extensions including CCCP with convex constraints. The authors also consider a more intricate extension of CCCP involving DC constraints. To accommodate this, the authors develop an extension of FW -- called FW+ -- for dealing with concave minimization subject to concave <= constraints. Applying this algorithm to the appropriate epigraph reformulation of CCCP with DC constraints yields a natural extension of CCCP, called CCCP+, with similar non-asymptotic convergence guarantees as before.

**Questions:**

Despite the strengths listed above, still some steps can be taken to improve the quality of the paper. I outline my main comments/questions below.

1. **Being more upfront about assumptions:** There are several implicit assumptions made in the paper (to my knowledge) that the authors should be more upfront about. I list these implicit assumptions below.
     - $f^* - g^*$, which is the optimal value of the DC problem, is assumed to be finite (hence not unbounded)
     - the above assumption implies that a solution to the CCCP subproblem always exists (authors should comment on this)
     - Related, the fact that $\phi^*$ is finite implies that the subproblem solutions always exist for FW in Section 2.1 (authors should comment on this)

2. **The stationarity criterion in Corollary 3.2** To be honest, it took me some time to realize that Corollary 3.2 is at all interesting. The main reason for this is that the authors did not motivate or discuss the stationarity criterion on the LHS of the inequality. The authors should add some discussion and be careful about doing this in the revised version. In particular, I would suggest defining a formal stationarity measure, e.g., $G(x_\tau)$ defined as the supremum of the LHS over all $x$. The authors should then add some discussion and perhaps a lemma justifying that this is an interesting stationarity criterion, i.e., the lemma should show that $G(x_\tau) = 0$ implies that $x_\tau$ is a stationary point of the DC problem. Such a lemma is very simple but not obvious to someone who is not readily familiar with DC programming. A result and some discussion along these lines -- currently there is none -- would strongly help to justify the value of Cor 3.2.

3. Why is the CCCP problem setting and algorithm defined twice in Sections 2.2 and 3? The discussion should be consolidated and moved to one location.

4. There seems to be a strong disconnect between CCCP+ and FW+ since FW+ deals with a *single* differentiable functional constraint and CCCP+ deals with multiple. This is not commented on as far as I can tell. Can FW+ be extended to multiple differentiable constraints so it can apply to CCCP+? (I suspect yes.)

5. The simplicity of the proofs of Lemma 2.1 and Proposition 3.1 suggest that there should be a simple and **direct proof** of Corrollary 3.2. Is there? While I greatly appreciate the proof based on the equivalence with FW -- indeed that is the main point of this paper and how Cor 3.2 came about -- still a direct proof may add some clarity to the ideas as it helps to cement everything and remove any doubts. I think that adding a short alternative direct proof to the supplement (or at least a discussion) would add expository value to the paper.

6. The authors say several times in several places that the analysis of FW considered -- in particular for concave minimization -- does not rely on the typical bounded curvature/Lipschitz smoothness assumptions. Another surprising fact is that it does not rely on a *bounded domain* assumption. (In a sense, this is included in the curvature part but is only obvious to an expert.) I would also add emphasis to this point, since in particular the domain for CCCP+ will be an unbounded epigraph, which is a non-traditional aspect of FW as well.

Minor comments:

The paper is quite well-written and I did not find many typos. Still, I have the following minor comments:

1. Line 56:  "alternative" -> "alternatives"
2. Lemma 2.1 and line 99:  clarify in the statement of the Lemma that this is FW *with line-search*
3. Line 111:  The acronym "MM" has yet to be specified
4. Line 132:  The fact that KKT is necessary also requires a constraint qualification. In this case, Slater's condition will do.
5. Line 226:  "formula" should be plural

**Limitations:**

The authors have discussed several limitations in Section 5 -- including e.g., that the results do not work directly for non-smooth problems -- and have appropriately left these extensions for future works. Discussion concerning potential negative societal impact is not applicable for this paper.

**Strengths And Weaknesses:**

### Strengths

1. The main idea of this paper -- finding the right way to represent the DC problem so that CCCP is FW -- is novel, clear, and significant. On the surface it feels like a simple idea, but in my opinion this is a strength of the paper. The authors have cleverly uncovered a gap in the literature. To the best of my knowledge, the connection uncovered here is completely new.

2. DC programming has emerged as a convenient class for modeling certain structured non-convex problems, yet the analysis of algorithms for DC programming is not nearly as developed as convex optimization. DC programming -- and also FW -- are quite important topics in the optimization literature with strong connections to ML. This paper takes a strong step forward in our understanding of the convergence and structure of DC algorithms.

3. The direct implications of the equivalence are immediately developed by the authors. However, this paper has potential for longer lasting impact and may end up being quite significant. The connection between FW and CCCP uncovered allows the rich literature of FW analysis (which is mainly finite iteration convergence analysis) to be transferred to the DC literature (which is mainly asymptotic analysis). Ideas from DC literature may also have impact on FW. Additionally, as CCCP is related to others algorithms such as EM, there are potentials for even broader reaching impact as well. Quite simply the cross-cutting nature of this paper -- and it being the first such paper to draw a connection between FW and DC literatures -- makes it very significant to me.

4. The paper is well written and the proofs are easy to follow. Again, the latter is a strength of the paper to me. To the best of my knowledge the proofs are mathematically correct. The paper is simply a pleasure to read.

### Weaknesses

1. Although the paper is written very well, the authors are (in my opinion) not careful enough regarding certain assumptions/definitions. Some simple fixes (which I outline in the next section) can help dramatically to overcome this weakness.

2. Similar non-asymptotic convergence results for CCCP have appeared in [Abbaszadehpeivasti et al., 2021], which the authors point out in the supplement. Although this weakens the novelty of the implications of the connection, in my view this does not detract too much from the quality and novelty of this paper. The results of this paper are based on a novel structural connection between the two algorithms, and the proofs are very simple. These appealing connections and simple proofs are a strength not shared by [Abbaszadehpeivasti et al., 2021].

3. There are no numerical experiments. However, in my view this paper does not need them. They would be besides the main point of the paper.

---

> ### Author Response · Authors · 2022-08-02
> **Thank you for your feedback.**
>
> Thank you for carefully reading our draft, and for your detailed review with constructive comments and encouraging words. We are delighted that you found our paper `a pleasure to read'.
>
> ### Implicit assumptions
> We admit that these assumptions should have been listed explicitly and discussed in more detail. This even misled some other reviewers into thinking we were using a modified version of the FW algorithm, whereas we are not. In particular, the DC problem is assumed to have a finite solution so that the CCCP subproblems are well-defined (which appears as an implicit assumption in our paper, since we assume the CCCP subproblems admit a solution). This requirement can be transferred to the equivalent FW formulation as $\phi^*$ is finite, so the FW subproblems have solutions too. In our revision, we will elaborate more on these technical details and their relation to the bounded domain assumptions common in the FW literature.
>
> ### Stationarity criterion in Corollary 3.2
> Thank you for bringing up this point. In recent years many researchers (including us) have used these types of "FW-gap functions" to judge the performance of FW procedures, and we missed the fact that its significance might be less evident to other researchers. To improve our presentation, we will explain the significance of the stationarity condition and the motivation behind it in our revision.
>
> ### Sections 2.2. and 3
> Section 2.2 provides background on the known results for CCCP, and Section 3 establishes the connections between CCCP and FW. We made these sections separate to distinguish between what is known and what is new. In the revision, we will work further on the presentation and ensure that these sections are not repetitive.
>
> ### FW+ with multiple functional constraints
> This omission is simply a mistake in our presentation. All of our results on FW+, including the stationarity condition in Lemma 4.2, hold for problems with multiple functional constraints. We will fix this in our revision and apologize for the confusion caused.
>
> ### Direct proof for CCCP
> Yes, we can write a simple and direct proof for CCCP, similar to the proof of FW. We will present it in the supplement.
>
> ### Bounded domain assumptions
> This is a subtle but conceptually important aspect of our results. Interestingly, we can incorporate the bounded domain assumptions into the bounded curvature and finite solution conditions. This point is also crucial for the connections between FW and proximal point method, mirror descent, and mirror-prox that we have included in the appendices (as wider implications of the epigraphical viewpoint introduced in the paper). We will elaborate more in this in our revision.
>
> ### Existing non-asymptotic convergence guarantees on CCCP
> We are grateful for the acknowledgment of our conceptual contribution. We included the discussion on [Abbaszadehpeivasti et al., 2021] in the supplementary material because we only discovered that work after the submission deadline for the main paper. We will move it to the main part in our revision along with the correct context as currently noted in the appendix.
>
> ### Lack of numerical experiments
> The focus of this paper is on the connections between FW/CCCP and FW+/CCCP+, and presenting numerical experiments on FW+ might distract the readers. We agree, however, that FW+ deserves further investigation and we left it for a follow-up work. There, we also plan to test and present its empirical performance.
>
> Remark / conceptual view
> ---
> The FW+ method advanced in the paper definitely merits further study, because it operates by linearizing not only the objective as vanilla FW does, but it also linearizes constraints at each iteration, which is a natural idea whenever constraints admit an easy linearization.

---

### Meta-Review · Area_Chair_qGLa · 2022-08-22

**Recommendation:** Accept
**Confidence:** Certain

**Metareview:**

Many reviewers have found that the paper contains many interesting elements and its theoretical derivations are solid. The paper is also very well written. There are some minor suggestions that could greatly help improve the quality of the final paper.

**Award:**

No

---

### Decision · Program_Chairs · 2022-09-14

Accept